# Agricultural production and air pollution: An investigation on crop straw fires

**Kai Zhao**[1], **Xiaohui Tian**[2]*, **Wangyang Lai**[3], **Shuai Xu**[2]

**1** Innovative Team in Agriculture-husbandry Economics, Institute of Animal Science, Chinese Academy of Agricultural Sciences, Beijing, China, **2** School of Agricultural Economics and Rural Development, Renmin University of China, Beijing, China, **3** School of Economics, Shanghai University of Finance and Economics, Shanghai, China

* 573486318@qq.com

**Data Availability Statement:** The data underlying the results presented in the study are available from National Aeronautics and Space Administration (NASA) and the National Oceanic

## Abstract

In numerous developing nations, the pervasive practice of crop residue incineration is a principal contributor to atmospheric contamination in agricultural operations. This study examines the repercussions of such biomass combustion on air quality during the autumnal harvest season, utilizing data acquired from satellite-based remote sensing of fire events and air pollution measurements. Employing wind direction information alongside difference-in-difference and fixed-effects methodologies, this investigation rectifies estimation inaccuracies stemming from the non-random distribution of combustion occurrences. The empirical findings reveal that agricultural residue burning precipitates an elevation in average PM2.5 and PM10 concentrations by approximately 27 and 22 µg/m$^3$ during the autumnal incineration period, respectively. Furthermore, air pollution attributed to residue burning in prominent grain-producing regions exceeds the national average by approximately 40%. By integrating economic paradigms into agri-environmental inquiries, this study offers novel insights and substantiation of the environmental expenditures engendered by crop residue burning, juxtaposed with extant meteorological and ecological research findings.

## Introduction

In numerous developing nations, the pervasive incineration of agricultural residue constitutes a critical contributor to atmospheric contamination. For instance, in China, an estimated 200 million metric tons of crop waste are discarded, comprising 17–22% of the total straw output [1]. The combustion of such organic matter not only squanders valuable resources but also engenders considerable environmental degradation [2]. Crop residue burning releases copious amounts of particulate matter (PM2.5 and PM10) and pernicious gases, including CO, SO2, NOx, and VOCs [3].

Recent investigations have implicated straw combustion as a primary instigator of severe regional haze pollution incidents, posing significant health hazards to local populations and impeding economic development [4, 5]. Consequently, quantifying the environmental impact of crop residue burning is imperative, furnishing a scientific foundation for determining the marginal consequences of regulatory measures. To date, economic research concerning straw

and Atmospheric Administration Integrated Surface Database (NOAA-ISD).

**Funding:** National Natural Science Foundation of China (NSFC) 71603268&71803119. The funders had no role in study design, data collection and analysis, decision to publish, or preparation of the manuscript.

**Competing interests:** The authors have declared that no competing interests exist.

burning and air pollution remains scarce, with prevailing studies predominantly concentrating on two primary domains: environmental ecology and atmospheric meteorology. These investigations encompass characterizations of pollutant emission factors, emission inventory calculations [6–10], and correlations with air pollution events [11–13]. A consensus has emerged, asserting that straw combustion exerts a profound influence on air quality, exhibiting pronounced temporal and spatial clustering and heterogeneity.

Nevertheless, extant research merely establishes correlations between air pollution and straw burning, neglecting to substantiate causal relationships. For instance, studies examining pollutant emission factors and emission inventory calculation methods fail to disentangle the confounding influence of economic variables on crop residue burning-induced pollution. Correlational analyses of air pollution events are constrained to specific instances in particular regions, yielding non-generalizable conclusions. Furthermore, most investigations center on the ramifications of straw burning-induced air pollution within urban environments, neglecting rural regions as crucial sources of contamination [14]. Moreover, these studies typically rely on annualized data at the provincial or city level, rendering it challenging to discern the seasonal attributes of straw burning practices.

This study scrutinizes the atmospheric repercussions of crop residue combustion during the autumnal harvest season, leveraging weekly satellite-derived fire point, air pollution, and agricultural economic data at the county level for the 2018–2019 period. This research employs several empirical identification strategies to infer the causal environmental effects of straw burning in China. Firstly, a fixed-effects model is utilized to account for unobserved heterogeneity in time-invariant factors. Secondly, a difference-in-difference approach is adopted to establish causal connections, designating counties with below-average fire points as the control group and those with above-average fire points as the treatment group. Non-harvest seasons serve as non-treatment periods, while harvest seasons constitute treatment periods. Lastly, the stochastic nature of wind direction is employed to assess the robustness of the difference-in-difference model results, effectively eliminating errors attributable to omitted explanatory variables. Consequently, this investigation not only elucidates the causal relationship between straw burning and air pollution–a lacuna in environmental ecology and atmospheric meteorology research–but also surmounts endogeneity concerns inherent in empirical economic inquiries.

The primary contributions of this study are as follows: First, it offers precise estimations of the air pollution consequences of straw burning at a national level, serving as a vital extension of the air pollution literature and a necessary augmentation of research on agricultural non-point source pollution. Second, it effectively discerns the causal relationship between straw burning and air pollution, addressing the endogeneity issue. The conclusions demonstrate that, by employing an economic analysis paradigm and comparing it with the results from natural science paradigms, current meteorological and ecological scientific research findings The conclusions demonstrate that, by employing an economic analysis paradigm and comparing it with the results from natural science paradigms, current meteorological and ecological scientific research findings [11, 14, 15] may have different perspectives on the environmental costs engendered by crop residue combustion.

The remainder of this study is organized as follows: The "Research Design" section outlines the theoretical framework, including modeling approaches and data processing techniques. The "Empirical Study on Straw Burning and Air Quality" section discusses empirical findings, robustness checks, and analyses of heterogeneity. Finally, the "Conclusions and Discussion" section provides concluding thoughts and reflections on the implications of the findings.

## Research design

### Data source

This study assembles data on air quality, straw burning, meteorological conditions, and county-level economic factors. Initially, the correlation between straw burning and air pollution is analyzed, followed by an examination of the impact of straw burning on air pollution, utilizing meteorological and regional economic data as control variables. The data set sources and brief descriptions are as follows:

The air quality dataset is derived from official statistics provided by the Ministry of Environmental Protection of China, which has been extensively employed in economic research [5, 16, 17]. The dataset encompasses daily air quality information (e.g., PM2.5 and PM10) from 2018 to 2019, collected from 1,497 air monitoring stations distributed throughout China. Monitoring station selection is predicated on three criteria: (1) exclusion of stations proximal to industrial pollution sources and major traffic routes. This criterion aimed to mitigate the influence of localized pollution sources that could skew the air quality data related to crop residue burning. Stations near industrial areas or major roads are likely to report higher levels of particulate matter due to emissions from vehicles and industrial activities. By excluding these stations, we sought to isolate the impact of crop residue burning on air quality from other anthropogenic sources of air pollution.(2) avoidance of horizontal obstacles exceeding the monitoring station.The presence of significant horizontal obstacles, such as tall buildings or natural formations, can alter local airflow patterns and potentially trap pollutants, leading to atypical readings at the monitoring station. To minimize these effects, we selected stations with minimal obstruction in their immediate vicinity, ensuring that our data more accurately reflects broader air quality conditions rather than localized anomalies. (3) minimization of terrain and weather effects in the local region.(3)Terrain features (e.g., valleys, mountains) and local weather patterns (e.g., persistent fog conditions) can dramatically influence air quality measurements by affecting pollutant dispersion and concentration levels. Stations in areas heavily influenced by such factors were excluded to prevent these natural phenomena from confounding the effects of crop residue burning on air quality. By adhering to these criteria, we aimed to curate a dataset that more accurately represents the impact of crop residue burning across different regions, thereby enhancing the reliability and validity of our findings. This meticulous approach to station selection underscores our commitment to producing robust and meaningful insights into the environmental costs of agricultural practices.

The straw burning dataset is obtained from the National Aeronautics and Space Administration (NASA) satellite's remote sensing data. NASA's fire resource management system, which employs the Moderate Resolution Imaging Spectroradiometer (MODIS), is capable of detecting fire points within an area of one hectare per hour. The identification of fire points relies on spotting anomalies within a pixel (250 square meters) by using a contextual algorithm that leverages the mid-infrared radiation emitted by fires. According to extant research, this dataset can reasonably and accurately gauge regional straw burning activity [1, 4, 18]. The number of fire points is selected over the extent of fire areas due to the nature of China's small-scale and fragmented agricultural economy. Aggregating small land plots from various farmers within fire areas does not accurately portray the true severity of straw burning. Smaller fires may be overlooked, but they generally contribute less to air pollution. If we were to simply sum up the number of large and small fires to obtain a total number of fire points, it would introduce measuring error. Thus, our focus is more on the larger fires which have a more significant impact on air pollution. Moreover, this study primarily focuses on the burning of crop straw rather than crop stubble. This is due to several reasons. Firstly, stubble burning typically does not lead to widespread fires or significant air pollution when compared to straw burning.

Secondly, the practice of individually igniting stubble is both labor-intensive and economically inefficient. As a result, stubble burning is relatively uncommon in China. Besides, we opted for the MODIS dataset over the VIIRS data, despite the latter's higher spatial resolution, due to considerations aligned with our research scope and objectives. The MODIS dataset's extensive historical record and consistent detection capabilities across different times of day made it more suitable for analyzing large-scale fire dynamics over long periods, which was essential for our study's focus. Additionally, the integration of VIIRS would have required significant pre-processing to ensure data compatibility, a task that exceeded our project's resource constraints. This decision was made to ensure continuity, reduce potential biases, and uphold the scientific integrity of our analysis without compromising our study's goals.

Meteorological data is sourced from the National Oceanic and Atmospheric Administration Integrated Surface Database (NOAA-ISD). This dataset comprises data from 407 meteorological monitoring stations in China, operational from 2018 to 2019. Meteorological indicators include temperature, dew point, precipitation, wind speed, and wind direction.

This study integrate economic variables to enhance our analysis of the impact of crop residue burning on air quality. Specifically, we include quarterly city-level Gross National Product (GDP) and monthly city-level industrial added value data. Industrial added value represents the net output of all industrial sectors, calculated as the difference between the gross output value of industrial production and the input value of the production process, such as raw materials and labor costs. This measure provides insights into the economic productivity of industrial sectors, excluding external factors like inflation. For this analysis, the data on GDP and industrial added value were sourced from the National Bureau of Statistics of China, which regularly publishes comprehensive economic data. These statistics are recognized for their reliability and are widely used in economic research to assess the development and performance of the Chinese economy. By incorporating these economic indicators into our empirical model as control variables, we aim to account for the influence of economic development on air quality, thereby providing a more nuanced understanding of the relationship between agricultural practices, economic conditions, and environmental outcomes.

Ultimately, air monitoring stations, meteorological monitoring stations, and fire point locations are matched to the nearest county based on their longitude and latitude coordinates to compile county-level data. Concurrently, the data are adjusted to the weekly county level, with the exception of GDP and industrial added value data, as demonstrated in Table 1 of the descriptive statistics provided below:

Furthermore, this study highlights the seasonality and evolving trends of straw burning. Fig 1 illustrates the average weekly fire points from 2001 to 2019, exhibiting a cyclical pattern that corresponds to the seasonality of agricultural production. The figure reveals three notable

**Table 1. Descriptive statistics.**

| Variables | Observations | Mean | Standard Deviation | Min | Max |
|---|---|---|---|---|---|
| PM2.5 (mcg/m$^3$) | 117,443 | 48.79 | 36.25 | 1.95 | 473.74 |
| PM10 (mcg/m$^3$) | 117,362 | 80.60 | 52.01 | 6.18 | 638.91 |
| Straw burning fire points (count number) | 118,726 | 0.85 | 7.19 | 0 | 581 |
| Temperature (˚C) | 117,871 | 15.88 | 10.49 | −31.48 | 34.05 |
| Dew point temperature (˚C) | 117,871 | 10.63 | 11.02 | −35.37 | 28.99 |
| Precipitation (mm) | 117,872 | 28.18 | 49.95 | 0 | 1140 |
| Windspeed (m/s) | 117,862 | 2.32 | 0.97 | 0.43 | 12.44 |
| GDP (hundred trillion RMB) | 62,261 | 8.92 | 9.22 | 0.77 | 57.24 |
| Industrial added value (trillion RMB) | 31,791 | 12.57 | 11.91 | 0.30 | 86.40 |

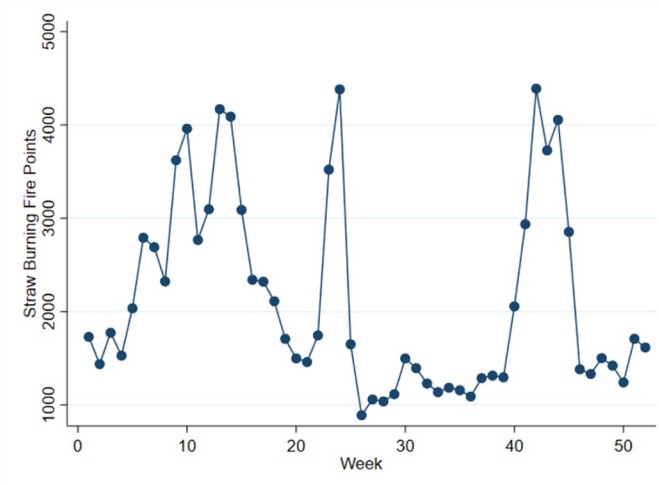

**Fig 1. Average weekly straw burning fire points in 2001–2019.**

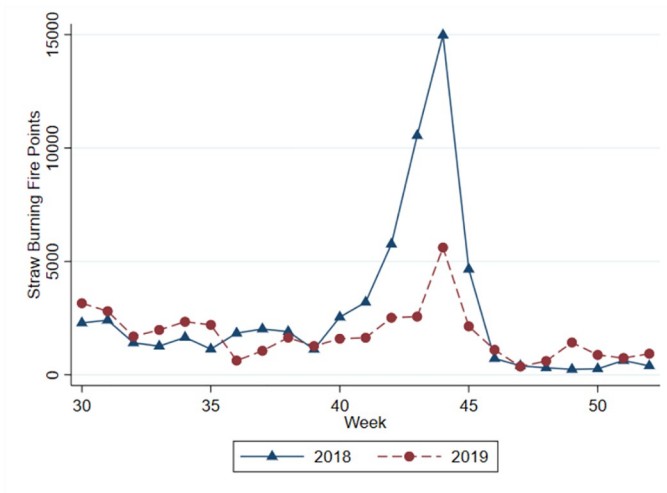

**Fig 2. National weekly straw burning fire point in 2018 and 2019.**

growth trends in fire points: firstly, during the Chinese Lunar New Year, fire points surge in the tenth week due to fireworks displays. Secondly, fire points experience an increase during the summer harvest season around the twenty-fourth week. Lastly, the apex of the straw burning period occurs during the autumn harvest season, around week 45 [19]. Fire points during the harvest season quadruple in comparison to the non-harvest season. This study refrains from examining the air pollution effects of straw burning during the summer harvest season, as the majority of grain production in China stems from the autumn harvest season. Additionally, Fig 2 depicts straw burning fire points between weeks 30 and 52 in 2018 and 2019. Despite a significant reduction in China's straw burning fire points due to stricter regulations in 2019, the seasonal trend remains analogous [20].

## Model setting

To scrutinize the influence of straw burning on air quality, the subsequent panel regression model, Eq (1), can be formulated:

$$PM_{it} = \beta_0 + \beta_1 count_{it} + \beta_2 X_{it} + u_i + \lambda_t + \varepsilon_{it} \tag{1}$$

Where $PM_{it}$ represents the PM2.5 and PM10 at county i in week t; $count_{it}$ signifies the counted fire point at county i in week t; $X_{it}$ indicates the control variables for meteorological and economic conditions, which include weekly average temperature, dew point temperature, wind speed, and industrial added value. Additionally, $u_i$ represents the individual fixed effect by introducing dummy variables for each county to capture the impact of individual characteristics that remain constant over time; $\lambda_t$ denotes the time fixed effect by adding dummy variables for each week to account for time-varying effects such as seasonality and business cycles. In essence, incorporating individual and time fixed effects in panel data analysis can mitigate the endogeneity issue between independent and dependent variables while enhancing the model's accuracy and explanatory power. Concurrently, $\varepsilon_{it}$ symbolizes the error term of the multiple linear regression model.

A multiple linear regression model is employed to examine the influence of straw burning on air quality, as it facilitates the investigation of relationships between multiple variables, including straw burning and air quality, while controlling for other pertinent factors. Moreover, by incorporating different control variables and analyzing shifts in the core explanatory variable's coefficients, the study's robustness can be evaluated.

Building on Eq (1), we utilize a difference-in-d-difference (DID) model to examine the impact of straw burning on air quality. Specifically, counties with sub-average fire points are designated as the control group, while counties with above-average fire points form the treatment group. In situations where random grouping is not feasible, dividing groups based on the mean value of the core explanatory variable can ensure a discernible difference between the control and experimental groups in a quasi-experimental design. Simultaneously, we designate the non-harvest season and harvest season as the non-treatment period and treatment period, respectively. The net effect of straw burning on air quality can be calculated by $\beta_3$ in Eq (2) and Table 2:

$$PM_{it} = \beta_0 + \beta_1 fire_i + \beta_2 period_t + \beta_3 fire_i * period_t + \beta_4 X_{it} + u_i + \lambda_t + \varepsilon_{it} \tag{2}$$

Where $fire_i$ denotes the dummy variable for the treatment group ($fire_i = 1$) and the control group ($fire_i = 0$); $period\_t$ signifies the dummy variable for the treatment period ($period_t = 1$) and the non-treatment period ($period_t = 0$).

## The empirical study of straw burning and air quality

### Regression results

Table 3 presents the regression results for the impact of straw burning fire points on PM2.5 and PM10 indexes. To investigate changes in the coefficients of the core explanatory variable,

**Table 2. DID model.**

|  | Non-Treatment Period (Period$_t$ = 0) | Treatment Period (Period$_t$ = 1) | Difference |
|---|---|---|---|
| Treatment group(fire$_i$ = 1) | $\beta_0 + \beta_1$ | $\beta_0 + \beta_1 + \beta_2 + \beta_3$ | $\beta_2 + \beta_3$ |
| Control group(fire$_i$ = 0) | $\beta_0$ | $\beta_0 + \beta_2$ | $\beta_2$ |
| DID | $\beta_1$ | $\beta_1 + \beta_3$ | $\beta_3$ |

**Table 3. The influence of straw burning point on PM index in panel regression model.**

| Explanatory Variables | Explained variables | | | | | |
|---|---|---|---|---|---|---|
| | PM2.5 | | | PM10 | | |
| Fire | 0.313 *** | 0.306 *** | 0.449 *** | 0.372 *** | 0.323 *** | 0.505 *** |
| | (0.039) | (0.039) | (0.097) | (0.045) | (0.041) | (0.094) |
| Temperature | | −0.522 *** | −0.569 *** | | 1.077 *** | 0.340 ** |
| | | (0.080) | (0.177) | | (0.144) | (0.238) |
| Dew | | −1.018 *** | −1.567 *** | | −2.635 *** | −2.767 *** |
| | | (0.062) | (0.172) | | (0.088) | (0.225) |
| Precipitation | | −0.037 *** | −0.032 *** | | −0.047 *** | −0.043 *** |
| | | (0.002) | (0.002) | | (0.002) | (0.003) |
| Windspeed | | −4.269 *** | −3.207 *** | | −7.348 *** | −4.637 *** |
| | | (0.232) | (0.344) | | (0.342) | (0.480) |
| GDP | | | 0.476 ** | | | 0.100 |
| | | | (0.194) | | | (0.272) |
| Industry added Value | | | −0.144 * | | | −0.116 |
| | | | (0.078) | | | (0.114) |
| Obs | 117,443 | 116,579 | 22,582 | 117,362 | 116,498 | 22,575 |
| $R^2$ | 0.41 | 0.45 | 0.51 | 0.40 | 0.44 | 0.49 |
| Time and individual fixed effect | √ | √ | √ | √ | √ | √ |

Country level clustered robust standard errors in parentheses.

*** $p < 0.01$,

** $p < 0.05$,

* $p < 0.1$.

we employ the stepwise regression method. According to the results, when adding control variables, the fire point of straw burning exerts a statistically significant positive impact on PM2.5 and PM10, with coefficients of 0.45 and 0.51, respectively. In other words, a one-unit increase in straw burning fire leads to a 0.45 and 0.51 increase in PM2.5 and PM10, respectively, holding all else constant. These results align with He, Liu and Zhou [4] but exhibit smaller coefficients than those found by Rangel and Vogl [21]. The discrepancy may arise from systematic differences in environmental pollution between China and Brazil, such as crop planting structures and climatic and geographic conditions. Nonetheless, the two-way fixed effects panel regression model cannot address endogeneity problems stemming from measurement errors and omitted variables. Consequently, we employ the DID method to further explore the relationship between straw burning and air pollution.

Compared to the results of the two-way fixed effects panel regression model, the DID model focuses more on the increased air pollution during the autumn harvest. Table 4 presents the regression results of straw burning's net impact on the PM2.5 and PM10 indexes. Controlling for meteorological conditions and local economic conditions, the average treatment effect of straw burning behavior on the PM2.5 and PM10 indexes is approximately 27 and 22 mcg/m³ during the autumn straw burning period. In the descriptive statistics, the average values of PM2.5 and PM10 are 48 and 80, respectively. Therefore, it is estimated that the contribution rate of autumn straw burning to the PM2.5 and PM10 indexes is 56% and 28%. Moreover, the impact of straw burning on PM2.5 is higher than on PM10, since biomass burning does not generate large particles, which is consistent with Jiang, Huo [14]. Additionally, $R^2$, the goodness of fit in the linear regression, can be further improved by adding more control variables.

**Table 4. The influence of straw burning point on PM index in DID model.**

| Explanatory Variables | PM2.5 | | | PM10 | | |
|---|---|---|---|---|---|---|
| | (1) | (2) | (3) | (4) | (5) | (6) |
| Treatment effect (fire*period) | 26.853 *** | 18.023 *** | 26.867 *** | 28.775 *** | 16.719 *** | 21.603 *** |
| | (1.310) | (1.194) | (2.956) | (1.457) | (1.363) | (3.357) |
| Temperature | | −0.455 *** | −0.568 *** | | 1.154 *** | 0.400 ** |
| | | (0.077) | (0.170) | | (0.112) | (0.233) |
| Dew | | −0.909 *** | −1.306 *** | | −2.549 *** | −2.625 *** |
| | | (0.061) | (0.177) | | (0.088) | (0.232) |
| Precipitation | | −0.036 *** | −0.031 *** | | −0.046 *** | −0.042 *** |
| | | (0.002) | (0.002) | | (0.002) | (0.003) |
| Windspeed | | −4.153 *** | −3.256 *** | | −7.218 *** | −4.615 *** |
| | | (0.231) | (0.322) | | (0.340) | (0.459) |
| GDP | | | 0.289 | | | −0.060 |
| | | | (0.188) | | | (0.266) |
| Industry added Value | | | −0.132 * | | | −0.085 |
| | | | (0.078) | | | (0.133) |
| Obs | 117,443 | 116,579 | 22,582 | 117,362 | 116,498 | 22,575 |
| $R^2$ | 0.42 | 0.45 | 0.51 | 0.41 | 0.44 | 0.49 |
| Time and individual fixed effect | √ | √ | √ | √ | √ | √ |

Country level clustered robust standard errors in parentheses.

*** $p < 0.01$,

** $p < 0.05$,

* $p < 0.1$.

However, high-frequency county-level data is severely limited, and adding more control variables could lead to collinearity issues.

Compared to previous meteorological and ecological scientific research, the air pollution effect of straw burning in our results is higher. For instance, Jiang, Huo [14] found that the annual average contribution of open-air straw burning on PM2.5 in Beijing, Dongying, and Chengdu was 12%, 15.8%, and 11%, respectively. Cheng, Wang [11] discovered that the contribution of open-air straw burning during the 2011 grain harvest to PM2.5 in the Yangtze River Delta cities was as high as 37%. Yu, Wang [15] determined that the contribution of open-air straw burning to PM2.5 in Beijing in the autumn, winter, and summer of 2010 was 19%, 25%, and 37%, respectively.

Since these meteorological and ecological scientific studies focus on correlation analysis instead of causal inference, their results may have sample selection or estimation bias. The higher regression coefficients in this study compared to meteorological and ecological scientific research suggest that the air pollution effect of straw burning might be underrated in previous studies. This highlights the importance of using robust econometric methods and causal inference techniques to obtain more accurate estimates of the impact of straw burning on air pollution.

## Robustness test

**Parallel trend test.** To address the requirement for a more comprehensive explanation regarding the variation of PM2.5 levels in 2019 during the harvest season between the treatment and control groups, we expand our analysis within the framework of the Difference-in-

Difference (DID) approach. This approach hinges on the parallel trends assumption, crucial for validating the causal relationships explored in our study.In 2019, a specific focus on PM2.5 variations during the harvest season reveals insightful dynamics between our designated treatment and control groups. While Fig 3, referencing PM2.5 data from 2018, illustrates the groups' adherence to parallel trends outside the harvest season, an analogous examination for 2019 underscores a significant deviation during the period of straw burning. This deviation is characterized by a marked increase in PM2.5 levels within the treatment group, which engaged in crop residue burning, as opposed to the control group, which did not.This differential trend during the harvest season of 2019 not only affirms the parallel trends assumption—by demonstrating comparable trajectories in PM2.5 levels between the groups outside of the treatment period—but also reinforces the causal impact of straw burning on air quality. The significant uptick in PM2.5 concentrations among the treatment group, juxtaposed with the control group's stability, provides strong evidence of the specific contribution of straw burning to deteriorating air quality.By meticulously analyzing the variation in PM2.5 levels and affirming the parallel trends assumption, our study bolsters confidence in the DID model's capacity to discern the true effects of agricultural practices on air pollution. This analytical rigor ensures that the observed differences in air quality between the treatment and control groups can be attributed with greater certainty to the act of straw burning, offering a solid foundation for causal inference within our economic analysis paradigm.

**Wind direction test.** Inasmuch as unobservable factors (e.g., local economic circumstances) may compromise the validity of regression outcomes, a spurious correlation between straw combustion and atmospheric purity may arise. The wind trajectory examination efficaciously eradicates inaccuracies engendered by unobservable determinants, given the stochastic and autonomous nature of wind patterns. Should straw incineration exert a substantial influence on air quality, upwind straw burning would inevitably impact downwind regions. This methodology has garnered widespread adoption in contemporary environmental and health economics investigations [22–24]. In accordance with the wind trajectory test schema, Fig 4 delineates upwind and downwind vicinities. Explicitly, when the airstream proceeds from region A to region B within a 45-degree range to the left and right of the line linking the absolute coordinates of regions A and B, region A is designated as the upwind area of region B, and region B as the downwind area of region A.

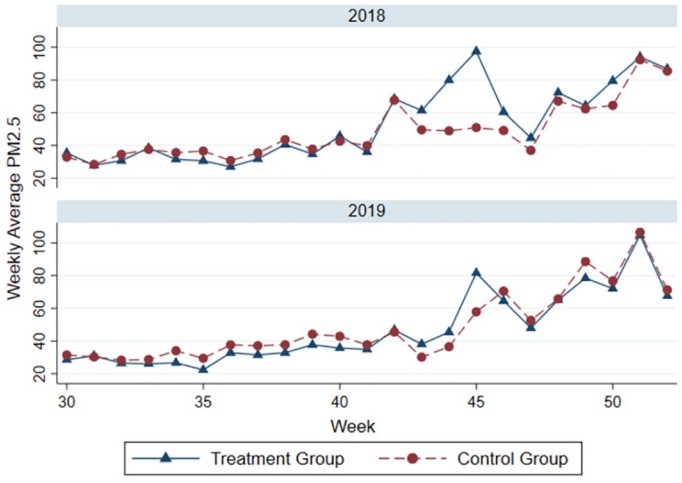

**Fig 3. Parallel trend test.**

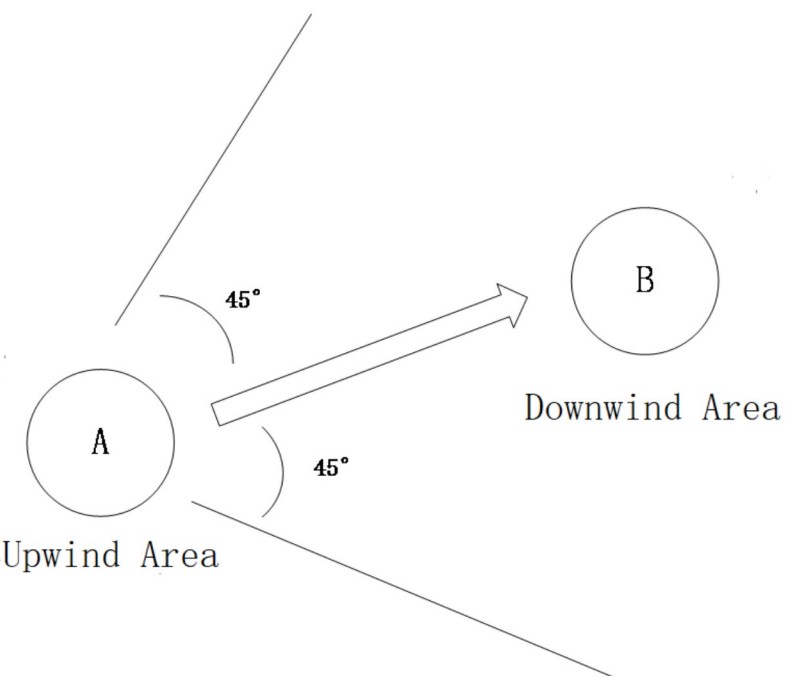

**Fig 4. Upwind and downwind areas.**

Fig 5 exhibits the processing of ignition point data for the Subsequent to the wind direction examination, Fig 5 portrays the ignition point data manipulation. The vector's magnitude and orientation embody wind velocity and inclination, correspondingly. The associated wind trajectory t3 represents the vector summation of vectors t1 and t2 across two temporal intervals. Building upon the aforementioned processes, upwind/downwind ignition point data within a 100 km radius surrounding the region are compiled.

Consequently, we scrutinize the impact of upwind/downwind ignition points on the area's air quality to execute a robustness assessment. In conjunction with Rangel and Vogl [21] method and the aforementioned DID model, the modified DID model (model 3) premised upon wind direction and ignition point is conceived as follows:

$$PM_{it} = \beta_0 + \beta_1 \text{upwindcount}_i * \text{period}_t + \beta_2 \text{downwindcount}_i * \text{period}_t + \beta_3 X_{it} + u_i + \lambda_t + \varepsilon_{it} \quad (3)$$

Where $\text{upwindcount}_i * \text{period}_t$ and $\text{downwindcount}_i * \text{period}_t$ denotes the treatment effect of upwind and downwind straw burning fire point respectively.

Moreover, the treatment group comprises regions where straw burning is prevalent during the harvest season, identified through satellite imagery as areas with a high number of ignition points. Conversely, the control group includes regions with minimal or no detected straw burning activity, serving as a baseline for comparison to understand the impact of straw burning on air quality. As evinced in Table 5, the upwind ignition point treatment effect retains statistical significance, thereby substantiating the robustness of the outcome. It is logically coherent for the treatment effect to be smaller than the coefficient displayed in Table 3, considering that the air quality monitoring station might not be situated within the county's confines. The transition from the initial, potentially biased regression results in Table 3 to the more robust and scientifically accurate estimations in Table 5, after addressing endogeneity and

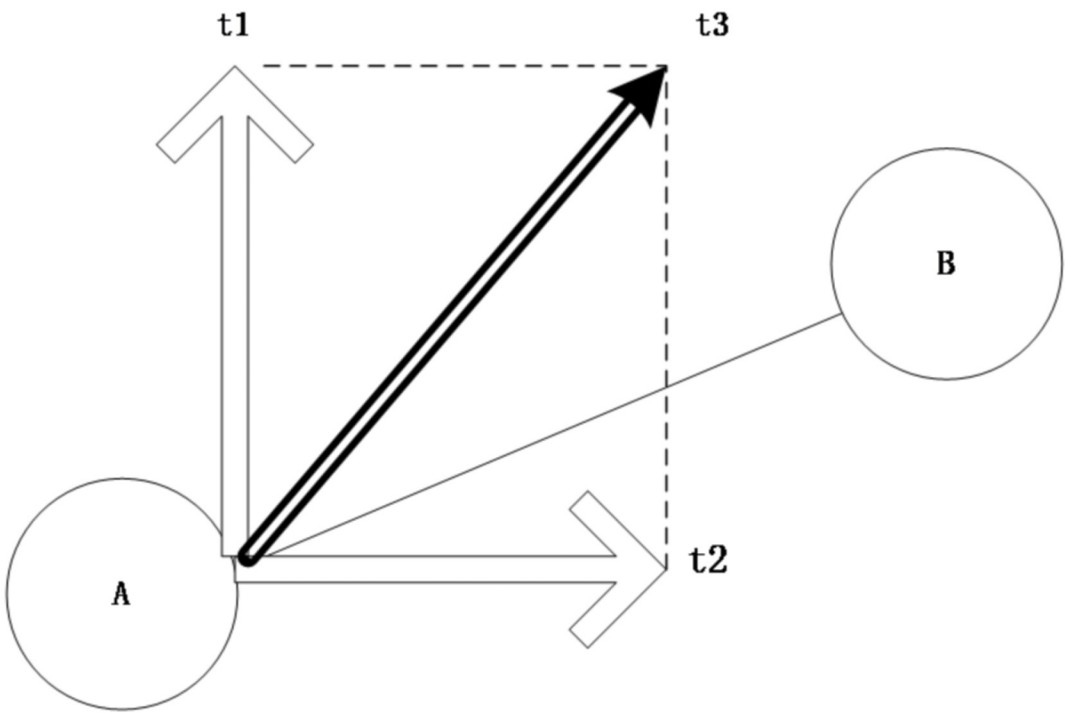

**Fig 5. Vector addition of wind.**

**Table 5. The effect of upwind/downwind straw burning on PM2.5 and PM10.**

| Explanatory Variables | PM2.5 | | PM10 | |
|---|---|---|---|---|
| | (1) | (2) | (3) | (4) |
| Upwind fire treatment effect (upwindfire*period) | 23.640 *** | 15.983 *** | 18.287 *** | 12.749 * |
| | (6.534) | (5.180) | (6.638) | (6.736) |
| Downwind fire treatment effect (downwindfire*period) | | 9.225 | | 6.668 |
| | | (7.018) | | (8.262) |
| Control variables | √ | √ | √ | √ |
| Obs | 22582 | 22582 | 22575 | 22575 |
| R2 | 0.509 | 0.510 | 0.490 | 0.490 |
| Time fixed effect and individual fixed effect | √ | √ | √ | √ |

City-level clustered robust standard errors are enclosed in parentheses (Given that not all air quality monitoring stations reside within the county, the robust standard deviation of clustering is more justifiable at the city level).

\*\*\* $p < 0.01$,

\*\* $p < 0.05$,

\* $p < 0.1$.

conducting robustness checks, highlights the evolution of our analysis from preliminary observations to refined causal inferences, enhancing the reliability and validity of our conclusions on the impact of straw burning on air quality. As a result, the impact on air quality is attenuated by distance. Additionally, the downwind treatment effect coefficient is diminutive and statistically insignificant, aligning with the intuitive notion that the downwind ignition point

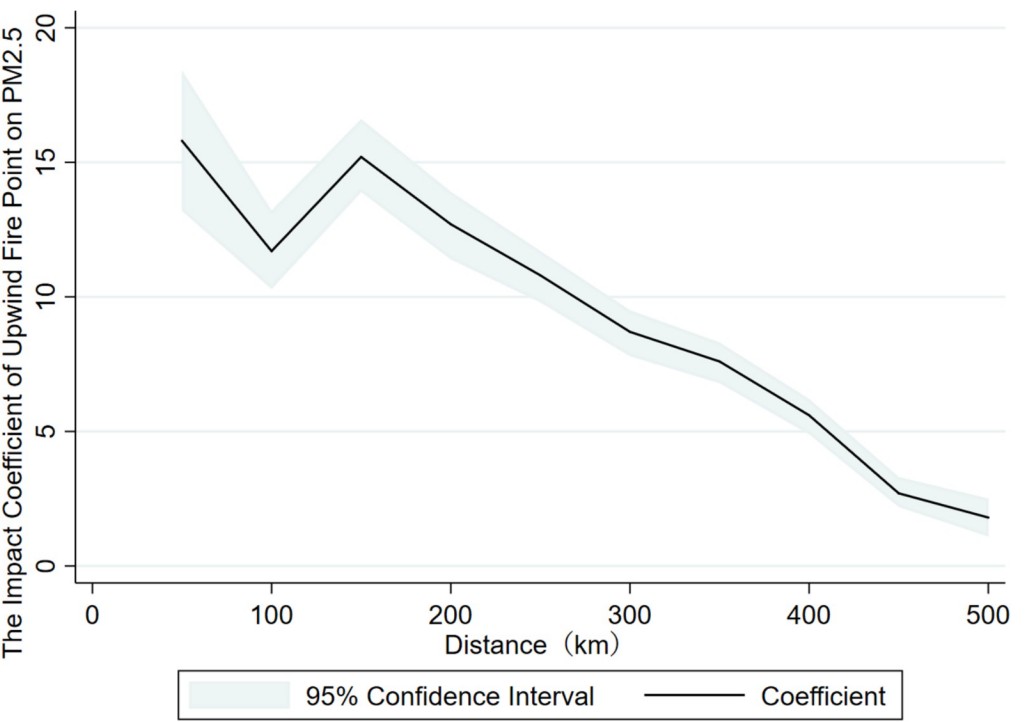

**Fig 6. Sensitivity test of fire points in different distances in upwind area.**

exerts a limited influence on air quality. This observation concurrently aligns with pertinent research [4, 21, 22]. It is crucial to underscore that the orographic features of the terrain also affect the results; however, as they remain constant over time, individual fixed-effect dummy variables duly account for them.

Fig 6 delineates the influence of upwind agricultural residue combustion on air quality at varying proximities surrounding a region. The coefficient of straw burning on air quality within the upwind vicinity exhibits a diminishing tendency as distance augments. The robustness examination, as previously alluded to, predicated on wind orientation, substantiates that straw burning exerts a substantial impact on air quality and possesses cross-regional consequences; its spatial spillover effect significantly attenuates as distance extends. Nevertheless, the coefficient undergoes aberrant oscillations between 50 and 150 km, attributable to the deliberate circumvention of pollution sources during the selection of pollution monitoring stations.

## Heterogeneity analysis

**Regional differences in pollution effects of straw burning.** Predominant grain-producing regions are characterized by favorable environmental conditions and relative advantages for cultivating primary food crops, encompassing thirteen provincial administrative jurisdictions (Liaoning, Hebei, Shandong, Jilin, Inner Mongolia, Jiangxi, Hunan, Sichuan, Henan, Hubei, Jiangsu, Anhui, Heilongjiang provinces). Projections indicate that during the autumnal harvest seasons of 2018 and 2019, the grain yield in these significant grain-producing regions constituted 76.37% and 76.05% of the aggregate national production, respectively. As corn, wheat, and rice are the principal crops cultivated in these regions, characterized by abundant

agricultural residue, a heightened prevalence of straw burning behavior may be anticipated in comparison to non-dominant grain-producing areas. Consequently, we shall investigate the disparity in air pollution engendered by straw burning between major and non-major grain-producing regions, as delineated in Table 6. The regression analysis reveals that straw burning activities during the autumn harvest season exert a positive influence on the PM2.5 index in the predominant grain-producing regions.

Moreover, according to model(3) of Table 6, the pollution effect of straw burning within major grain-producing regions is approximately 38%, which aligns with meteorological and ecological research such as Cheng, Wang [11], Jiang, Huo [14] and Yu, Wang [15]. Nonetheless, in the non-dominant grain-producing regions, the coefficients are either negligible or marginally negative. Although statistically significant, their economic relevance is inconsequential, suggesting that the ramifications of straw burning on the PM2.5 index are confined in these non-dominant grain-producing areas.

**Time differences in the pollution effect of straw burning.** In recent years, central and local authorities have enacted a series of policies pertaining to the prohibition and comprehensive utilization of agricultural residue to address the issue of straw burning. Additionally, as demonstrated in prior analyses, the incidence of straw burning during the 2019 autumn harvest has substantially diminished in comparison to 2018. Consequently, the net influence of straw burning on air pollution may exhibit variation. Table 7 presents the discrepancy in the impact of straw burning on the PM2.5 index between 2018 and 2019. According to the regression outcomes of the Difference-in-Differences (DID) model, the treatment effects of straw burning during the autumn harvest seasons of 2018 and 2019 are markedly significant. In

**Table 6. Regional differences in pollution effects of straw burning.**

| Explanatory Variables | Major Grain-Producing Areas | | | Non-Major Grain-Producing Areas | | |
|---|---|---|---|---|---|---|
| | (1) | (2) | (3) | (4) | (5) | (6) |
| Treatment effect (fire*period) | 31.257 *** | 27.406 *** | 38.203 *** | 1.212 | −2.562 | −8.461 |
| | (1.361) | (1.362) | (2.428) | (0.967) | (3.052) | (9.089) |
| Temperature | | 0.020 | −1.463 *** | | −0.543 *** | 0.299 |
| | | (0.118) | (0.285) | | (0.112) | (0.212) |
| Dew | | −0.452 *** | −0.219 | | −1.779 *** | −2.463 *** |
| | | (0.082) | (0.231) | | (0.102) | (0.274) |
| Precipitation | | −0.056 *** | −0.052 *** | | −0.019 *** | −0.014 *** |
| | | (0.002) | (0.004) | | (0.002) | (0.002) |
| Windspeed | | −2.673 *** | −1.745 *** | | −4.627 *** | −2.759 *** |
| | | (0.318) | (0.455) | | (0.342) | (0.476) |
| GDP | | | 0.356 | | | −0.560 ** |
| | | | (0.259) | | | (0.248) |
| Industry added value | | | 0.021 | | | 0.006 |
| | | | (0.104) | | | (0.061) |
| Obs | 71,481 | 71,386 | 12,504 | 45,962 | 45,193 | 10,078 |
| $R^2$ | 0.50 | 0.51 | 0.57 | 0.35 | 0.42 | 0.54 |
| Time and individual fixed effect | √ | √ | √ | √ | √ | √ |

Country level clustered robust standard errors in parentheses.

*** $p < 0.01$,

** $p < 0.05$,

* $p < 0.1$.

Table 7. Time differences in the pollution effect of straw burning.

| Explanatory Variables | 2018 | | | 2019 | | |
|---|---|---|---|---|---|---|
| | (1) | (2) | (3) | (4) | (5) | (6) |
| Treatment effect (fire*period) | 35.090 *** | 25.595 *** | 36.131 *** | 18.596 *** | 10.395 *** | 17.797 *** |
| | (1.756) | (1.593) | (3.902) | (1.106) | (1.067) | (2.707) |
| Temperature | | −0.312 *** | 0.229 | | −0.699 *** | −1.368 *** |
| | | (0.076) | (0.240) | | (0.103) | (0.168) |
| Dew | | −1.250 *** | −2.449 *** | | −0.547 *** | −0.026 |
| | | (0.077) | (0.249) | | (0.074) | (0.177) |
| Precipitation | | −0.050 *** | −0.044 *** | | −0.017 *** | −0.014 *** |
| | | (0.002) | (0.004) | | (0.002) | (0.002) |
| Windspeed | | −4.888 *** | −4.431 *** | | −3.315 *** | −1.477 *** |
| | | (0.259) | (0.404) | | (0.274) | (0.283) |
| GDP | | | 2.142 *** | | | 0.395 ** |
| | | | (0.430) | | | (0.194) |
| Industry added Value | | | −0.515 *** | | | −0.528 *** |
| | | | (0.160) | | | (0.078) |
| Obs | 58,813 | 58,311 | 12,310 | 58,630 | 58,268 | 10,272 |
| $R^2$ | 0.39 | 0.43 | 0.50 | 0.49 | 0.50 | 0.62 |
| Time and individual fixed effect | √ | √ | √ | √ | √ | √ |

Country level clustered robust standard errors in parentheses.

*** $p < 0.01$,

** $p < 0.05$,

* $p < 0.1$.

comparison to 2018, the air pollution effect of straw burning in 2019 declined by approximately 50%, signifying the efficacy of the pertinent policies.

## Conclusions and discussion

We undertake a multivariate regression analysis on Chinese county-level panel data (encompassing satellite-monitored agricultural residue combustion occurrences, air pollution records, and meteorological data) to examine the influence of straw burning behavior on air quality during the autumnal harvest season. Specifically, we employ the Difference-in-Differences (DID) model to scrutinize the net air pollution ramifications of straw burning, and deliberate on regional and temporal disparities.

Our findings reveal that straw burning during the autumn harvest season exerts a significantly positive impact on the PM2.5 and PM10 indices. Notably, throughout the autumnal straw burning interval, the PM2.5 and PM10 indices in areas with heightened straw burning were, on average, approximately 27 and 22 units higher (56% and 28%, respectively). The outcomes suggest that, through our economic analysis paradigm, we have arrived at insights that differ from those of contemporary meteorological and ecological scientific research regarding the environmental costs engendered by straw burning.

Furthermore, the air pollution effects of straw burning during the autumn harvest season are more pronounced in the predominant grain-producing regions. The escalation in the PM2.5 index attributable to straw burning is approximately 40% higher than the national mean. In non-dominant food production areas, the net air pollution effect of straw burning during the autumn harvest season is inconsequential. In comparison to 2018, the incidence of

straw burning during the 2019 autumn harvest season has been markedly reduced, with the PM2.5 index increase declining by roughly 50%. This indicates that the straw burning prohibition policy has effectively curtailed such behavior.

Drawing upon the findings of this study, we propose several policy recommendations. Crop residue represents one of the most abundant and accessible sources of biomass energy in rural China, rendering straw burning a significant squandering of resources. The substantial economic cost associated with farmers' utilization of straw is the primary impetus for this burning behavior [4]. Policies should initially concentrate on advancing the comprehensive use of agricultural residue and supporting the relevant industry. Simultaneously, the central government ought to offer enhanced support to major grain-producing regions and stimulate farmers' enthusiasm for straw recycling, as straw burning is more severe in these areas compared to other provinces. Moreover, the straw burning ban policy remains an effective tool for improving air quality; therefore, heightened supervision is imperative for controlling such behavior.

## Author Contributions

**Conceptualization:** Kai Zhao, Xiaohui Tian, Wangyang Lai, Shuai Xu.

**Data curation:** Kai Zhao, Shuai Xu.

**Formal analysis:** Kai Zhao, Shuai Xu.

**Funding acquisition:** Xiaohui Tian.

**Investigation:** Wangyang Lai, Shuai Xu.

**Methodology:** Xiaohui Tian, Wangyang Lai.

**Resources:** Shuai Xu.

**Software:** Kai Zhao, Shuai Xu.

**Supervision:** Kai Zhao, Xiaohui Tian, Wangyang Lai.

**Writing – original draft:** Shuai Xu.

**Writing – review & editing:** Kai Zhao, Shuai Xu.

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
