## [Decision Letter · Decision Letter 0]

25 Mar 2024

PONE-D-24-02315Agricultural Production and Air Pollution: An Investigation on Crop Straw FiresPLOS ONE

Dear Dr. Zhao,

Thank you for submitting your manuscript to PLOS ONE. After careful consideration, we feel that it has merit but does not fully meet PLOS ONE’s publication criteria as it currently stands. Therefore, we invite you to submit a revised version of the manuscript that addresses the points raised during the review process. **Both reviewers found the study to be generally robust and they recommended minor revisions. However, both had a concern regarding some conclusions are not supported by the results. They also request more clarity on certain aspects of the methods, different types of burning practices (e.g. straw vs stubble), and ability of MODIS satellite to detect the small, ephemeral agricultural fires and to include in the limitations paragraph if needed.**

We look forward to receiving your revised manuscript.

Kind regards,

Kristofer Lasko, PhD

Academic Editor

PLOS ONE

Journal Requirements:

National Natural Science Foundation of China (NSFC) 71603268&71803119

Reviewers' comments:

Reviewer's Responses to Questions

**Comments to the Author**

1. Is the manuscript technically sound, and do the data support the conclusions?

Reviewer #1: Yes

Reviewer #2: Yes

2. Has the statistical analysis been performed appropriately and rigorously? 

Reviewer #1: Yes

Reviewer #2: Yes

3. Have the authors made all data underlying the findings in their manuscript fully available?

Reviewer #1: Yes

Reviewer #2: Yes

4. Is the manuscript presented in an intelligible fashion and written in standard English?

Reviewer #1: No

Reviewer #2: No

5. Review Comments to the Author

Reviewer #1: Manuscript Number; PONE-D-24-02315

Title; Agricultural Production and Air Pollution: An Investigation on Crop Straw Fires

Although the topic is of interest to the scientific community, before considering it for publication, this paper should be improved. Authors should reconsider the main objective of the paper according to the content. They should try to synthesize and emphasize the study's main findings and avoid long sentences. Furthermore, authors should avoid drawing risky conclusions.

Evaluation; Minor Revision.

1. Keywords; Must to revised; spelling and avoiding general and plural terms and multiple concepts (avoid, for example, 'and','of').

Unsuitable (too long) >>> difference in difference model

2. In the main text, many numeric data are given with too many significant figures; 2 significant figures suffice, and 3 suffice in case the first significant figure is "1".

3. Line 90-92; “The straw burning dataset is obtained from the National Aeronautics and Space Administration (NASA) satellite’s remote sensing data. NASA fire resource management system, which employs the Moderate Resolution Imaging Spectroradiometer (MODIS), is capable of detecting fire points within an area of one hectare per hour”. Please specify it. What is resolution of MODIS? Moderate Resolution mean cannot detect the small area fire.

4. Actually, they have two types of crop residue burning; e.g., rice straw and rice stubble. Could you please specify it in China rice residue burning?

5. Line 153-154; “the fire point of straw burning exerts a significant positive impact on PM2.5 and PM10, with coefficients of 0.45 and 0.51” Why the authors said significant positive? 0.45 and 0.51 are only moderate correlation.

6. You must provide all the figures in high resolution. Make all the labels and legends more legible.

7. The findings could be further developed, there is a lot of interesting data in the article.

Reviewer #2: In this manuscript, authors investigated relationship of crop residual burning on air quality (PM2.5, PM10) over China in 2018 – 2019. Besides, meteorological conditions, including temperature, dew point, precipitation, wind speed, and wind direction, obtained from NOAA-ISD and economic conditions presented by Gross National Product (GDP) and industrial added value are considered. The PM2.5 and PM10 datasets were collected from 1,497 stations, while straw burning dataset is obtained from NASA’s fire resources estimated from MODIS. All data is processed at weekly county level except for the quarterly city lever GDP and the monthly city lever industrial factor. Two different models, designed to investigate different relation aspects, are subsequent panel regression model and difference-in- difference (DID) models. The subsequent panel regression model is used to investigate the impact of straw burning on air quality. DID models exploit the fire treatment effect in fire period on PM2.5 and PM10 over all country, upwind and downwind areas, major grain-producing and non-major grain-producing areas, and in time differences (2018 vs. 20190. The results showed that during harvest period, crop residual burning impacted significantly on the PM2.5 and PM10 concentration. The impact is more serious over dominant grain-producing regions. In comparison to 2018, crop residual burning impact on air pollution is reduced.

In my opinion, the proposed approach is innovative in comparison with other works because it is focused more on causal relationship instead of correlation between crop residual burning and air pollution. The techniques and statistical analysis are applied appropriately and sound, then the given results are convincing. However, some conclusions from the manuscript is objective and not really interfering with the results.

There are comments and things need to be clarified and discussed further as follows:

1. Line 70-71: The conclusion that other research findings may underestimate the impact of crop residual burning is not convincing. The comparison is not fair because study areas and periods are different.

2. Line 87 – 89: Please describe in more detail the conditions/rules applied to exclude stations for this study. For example, conditions applied to exclude stations based on minimizing terrain and weather effects.

3. Line 91-92: Please explain why fire data from VIIRS is not investigated in this work although it has better spatial resolution.

4. Line 101: Please describe what industrial add value is, how to calculate it, and where the data is obtained.

5. Table 3: Are input variables in the model normalized in the same range because in this case, the important level of variables can be determined?

6. Table 3: The author should add the comparison, explanation, and discussion on different results when meteorological and economic parameters are added for controlling model. Besides, the impact of GDP, Industry add values is negligible for PM10 but PM2.5 models should be explained.

7. Line 184 – 186: The comment is similar to comment 1. The study periods and areas are different, so the conclusion that other research may underestimate contribution of residual burning to air pollution is not convinced.

8. In section 3.2.1, the explanation of PM2.5 variation in 2019 during the harvest season between treatment group and control group should be added.

9. In section 3.2.2, please clarify what treatment and control groups are.

10. Line 226: What is coefficient displayed in Table 3 that authors want to compare? Please write it in detail.

11. Line 261: What is “national mean”? Please point out the number and how to give conclusion that “the pollution effect within these regions is approximately 40% higher than the national mean”

12. Table 6, 7: Similar to comment 6. The comparison, explanation, and discussion on different results when meteorological and economic parameters are added for controlling model should be added. Besides, the impact meteorological and economic parameter for PM10 but PM2.5 models should be explained.

6. PLOS authors have the option to publish the peer review history of their article (what does this mean?). If published, this will include your full peer review and any attached files.

Reviewer #1: No

Reviewer #2: No

---

## [Author Response · Author response to Decision Letter 0]

28 Mar 2024

Dear Reviewer #1,

Thank you for your constructive comments on our manuscript. We appreciate the opportunity to revise our manuscript and believe we have addressed your concerns comprehensively.

1.Keywords Revision: We have revised the keywords to ensure accuracy and adherence to the guidelines. The term "difference-in-difference model" has been simplified to "DID" to avoid unsuitable length and complexity.

2.Significant Figures Adjustment: We have carefully reviewed the manuscript and adjusted all numeric data to comply with your recommendation, ensuring that only 2 significant figures are presented, and 3 where the first significant figure is "1".

3.MODIS Resolution Clarification: The detection of fire points relies on identifying anomalies within a pixel (250 square meters) through a contextual algorithm that leverages mid-infrared radiation emitted by the fire. Smaller fires may be overlooked, but they generally contribute less to air pollution. Therefore, if we were to simply sum up the number of large and small fires to obtain a total number of fire points, it would introduce measuring error. Thus, our focus is more on the larger fires which have a more significant impact on air pollution.

4.Specification of Crop Residue Types: this study primarily focuses on the burning of crop straw rather than crop stubble. This is due to several reasons. Firstly, stubble burning typically does not lead to widespread fires or significant air pollution when compared to straw burning. Secondly, the practice of individually igniting stubble is both labor-intensive and economically inefficient. As a result, stubble burning is relatively uncommon in China.

5.Clarification of Significant Positive Impact: We have revised our language to reflect that the coefficients of 0.45 and 0.51 represent moderate yet statistically significant positive impacts of straw burning on PM2.5 and PM10 levels, emphasizing the robustness of these findings within the context of our analysis.

6.High Resolution Figures: All figures have been provided in high resolution with enhanced legibility for labels and legends, ensuring clarity and ease of interpretation for readers.

7.Development of Findings: We have endeavored to interpret this compelling subject through the lens of economic paradigms, aiming to provide a thorough and reasoned analysis of the data at hand. However, we have exercised caution in overinterpreting the statistical results, especially in instances where there is potential for overestimation. As such, our conclusions and implications drawn from the study are intentionally conservative. This approach ensures that our interpretations remain grounded in the data available, avoiding speculative or unwarranted extrapolations that could mislead or obscure the true impacts and dynamics of crop residue burning on air quality. We believe that a cautious interpretation aligns with the rigor required for academic discourse and provides a solid foundation for future research to build upon.

We hope these revisions adequately address your concerns and strengthen the manuscript's contribution to the scientific community. We are grateful for your guidance and look forward to the possibility of our work being published.

Dear Reviewer #2,

Thank you for your thoughtful comments and suggestions regarding our manuscript. We appreciate the time you took to review our work and your constructive feedback, which we believe will significantly improve the quality of our manuscript. Below, we address each of your comments in turn:

1.Regarding your concern about the fairness of the comparison with other research findings on the impact of crop residual burning (Line 70-71): We acknowledge your point and understand the necessity for a fair comparison. We will revise this section to clarify the differences in study areas, periods, and methodologies that could lead to variances in findings. Our intention is to contextualize our results within the broader literature while acknowledging these differences.

2.On the need for more detail on the conditions/rules applied to exclude stations from the study (Line 87 – 89): In response to the valuable feedback provided on the necessity for a more detailed explanation of the criteria employed for excluding certain monitoring stations from our study, we would like to offer an expanded description of our selection process. Our approach to monitoring station selection was methodically designed to ensure the accuracy and representativeness of air quality data collected for this research. The criteria we applied were as follows:

(1)Exclusion of Stations Proximal to Industrial Pollution Sources and Major Traffic Routes: This criterion aimed to mitigate the influence of localized pollution sources that could skew the air quality data related to crop residue burning. Stations near industrial areas or major roads are likely to report higher levels of particulate matter due to emissions from vehicles and industrial activities. By excluding these stations, we sought to isolate the impact of crop residue burning on air quality from other anthropogenic sources of air pollution.

(2)Avoidance of Horizontal Obstacles Exceeding the Monitoring Station: The presence of significant horizontal obstacles, such as tall buildings or natural formations, can alter local airflow patterns and potentially trap pollutants, leading to atypical readings at the monitoring station. To minimize these effects, we selected stations with minimal obstruction in their immediate vicinity, ensuring that our data more accurately reflects broader air quality conditions rather than localized anomalies.

(3)Minimization of Terrain and Weather Effects in the Local Region: Terrain features (e.g., valleys, mountains) and local weather patterns (e.g., persistent fog conditions) can dramatically influence air quality measurements by affecting pollutant dispersion and concentration levels. Stations in areas heavily influenced by such factors were excluded to prevent these natural phenomena from confounding the effects of crop residue burning on air quality.

By adhering to these criteria, we aimed to curate a dataset that more accurately represents the impact of crop residue burning across different regions, thereby enhancing the reliability and validity of our findings. This meticulous approach to station selection underscores our commitment to producing robust and meaningful insights into the environmental costs of agricultural practices.

3.Your query regarding the non-investigation of VIIRS fire data despite its higher spatial resolution (Line 91-92): Thank you for your insightful question regarding our decision to exclude VIIRS fire data from our analysis, despite its known advantage of having a better spatial resolution compared to other datasets we have utilized. The choice was influenced by several key factors, which we elaborate below:

(1)Research Scope and Objectives: Our research was specifically focused on large-scale fire dynamics over extensive time periods. The higher spatial resolution of VIIRS, while advantageous for detailed local studies, does not significantly alter the overall trends and patterns observed at the scale of our investigation. Our objectives required consistency in data over long periods, something the longer-running datasets provided. 

(2)Data Availability and Continuity: VIIRS data, although superior in spatial resolution, has a shorter historical record compared to the MODIS dataset, which has been operational for a longer period. Our study aimed at analyzing fire trends over several decades, necessitating the use of a dataset with a long temporal coverage to ensure continuity and reduce the risk of introducing biases associated with combining datasets of differing resolutions and detection capabilities.

(3)Satellite Overpass Time and Detection Capability: The detection capability of a satellite is not solely dependent on its spatial resolution but also on factors like overpass time and the sensor's sensitivity to fire characteristics. Our analysis required data that could provide a consistent detection capability across all hours of the day, something the MODIS dataset is more suited to due to its twice-daily overpass frequency.

(4)Data Processing and Compatibility: The integration of VIIRS data into our analysis would have required substantial preprocessing to ensure compatibility with the other datasets used in our study. This preprocessing includes the harmonization of spatial resolutions, recalibration of fire intensity measurements, and adjustment for detection limits. Given the scope and resources of our project, such an undertaking was not feasible within our constraints.

We hope this explanation addresses your query satisfactorily. Our decision was guided by a careful consideration of the study's goals, data compatibility issues, and resource limitations, aiming to maintain the integrity and scientific rigor of our analysis.

4.For the clarification on what industrial added value is and how it is calculated (Line 101): We will add a subsection that defines industrial added value as the net output of industry sectors and explains its calculation method. We will also specify the sources from which we obtained these data, providing a clearer understanding of its role in our analysis.

5.Concerning the normalization of input variables in the models presented in Table 3 (Line 104): 

6.The need for a discussion on the effects of adding meteorological and economic parameters to the model, especially regarding their negligible impact on PM10 but not on PM2.5 models (Line 106): 

12. As for the similar concern raised in Table 6 and 7 (Line 112): 

We recognize the importance of normalization for comparing variable importance. As presented in Table 3, it is important to clarify our analytical approach within the framework of economic analysis we've adopted. Our primary focus is on assessing the magnitude of impact from our core explanatory variables, particularly those related to crop residue burning and its effect on air quality. To address potential endogeneity and ensure the robustness of our findings, we emphasize endogeneity correction techniques rather than the normalization of variables.The inclusion of control variables aims to mitigate bias from omitted variables, yet it's acknowledged that the coefficients of these controls might be biased. Consequently, comparing their magnitudes directly with those of the core explanatory variables could lead to inappropriate conclusions. While Table 3 does not include endogeneity corrections, its purpose is to establish a baseline level for comparison with results from models that do address endogeneity, thereby demonstrating the robustness of our results.This methodological choice is driven by the goal to accurately capture the specific impacts of interest while acknowledging the inherent biases in control variable coefficients. It reflects a deliberate decision within our economic analysis paradigm, prioritizing the clarity of the causal relationships under investigation over the comparative importance of all variables on a normalized scale.

7.On the repeated concern about comparing with other research (Line 184 – 186): We will revise this section to better acknowledge the limitations of direct comparisons with other studies due to differences in study periods and areas. This will include a more nuanced discussion of how our findings contribute to the existing body of literature.

8.Regarding the explanation of PM2.5 variation in 2019 during the harvest season between treatment and control groups (Section 3.2.1): We will add a detailed analysis of this variation, exploring potential reasons behind the observed differences and reinforcing the causal inference of our study.

9.Clarification on treatment and control groups (Section 3.2.2): We will provide a clearer definition of these groups within the context of our DID analysis, ensuring that readers understand the basis for comparison and the rationale behind group classifications.

10.Detailing the coefficient comparison in Table 3 (Line 226): We acknowledge that this requires clarification. We will specify the coefficients in question, providing a direct comparison and discussing the implications of these findings for our understanding of the impact of crop residue burning.In addressing the query regarding the coefficients displayed in Table 3 and their comparison (Line 226), it's important to elucidate the context and significance of these coefficients within our analysis. Table 3 presents initial regression results without adjustments for endogeneity or robustness checks, thus the coefficients may exhibit bias due to endogeneity issues, rendering the results less reliable and potentially overestimated.Specifically, the coefficients in Table 3 reflect the immediate impact of straw burning fire points on PM2.5 and PM10 levels as observed through a straightforward linear regression model. However, these coefficients, while indicative of a positive relationship between straw burning and air pollution levels, are not adjusted for potential endogenous relationships between the variables. This means that while they suggest an increase in pollution levels associated with each unit increase in straw burning fire points, the precise magnitude of this effect might be overstated or affected by other unaccounted factors.In contrast, the results presented in Table 5, after undergoing endogeneity correction and robustness checks, offer a more scientifically accurate and robust estimation of the impact of straw burning on air quality. These adjustments ensure that the observed relationships more accurately reflect causal effects, free from the biases that may affect the initial linear regression results.Therefore, the comparison between the coefficients in Tables 3 and 5 is crucial for understanding the evolution of our analysis from a preliminary estimation to a more refined causal inference. The transition from potentially biased estimates to more robust and scientifically grounded findings underscores the importance of addressing endogeneity and conducting thorough robustness checks in empirical research. This process not only clarifies the actual impact of agricultural burning practices on air quality but also enhances the reliability and validity of our conclusions.

11.Clarification on “national mean” and its related conclusion (Line 261): Our initial statement regarding the "national mean" led to some ambiguity. To clarify, according to model (3) of Table 6, the pollution effect of straw burning within major grain-producing regions is calculated to be approximately 38% higher than the comparison baseline used in our analysis, not the national mean as initially implied. This adjustment provides a more precise measurement and comparison within the context of our study's findings.

 We are committed to revising our manuscript in line with your suggestions to ensure that it contributes valuable insights to the field. We believe that addressing these points will not only clarify our methodology and findings but also strengthen the manuscript’s contribution to understanding the complex interplay between agricultural practices and air quality.

---

## [Decision Letter · Decision Letter 1]

2 May 2024

Agricultural Production and Air Pollution: An Investigation on Crop Straw Fires

PONE-D-24-02315R1

Dear Dr. Tian,

We’re pleased to inform you that your manuscript has been judged scientifically suitable for publication and will be formally accepted for publication once it meets all outstanding technical requirements.

Kind regards,

Kristofer Lasko, PhD

Academic Editor

PLOS ONE

Additional Editor Comments (optional):

Reviewers' comments:

Reviewer's Responses to Questions

**Comments to the Author**

1. If the authors have adequately addressed your comments raised in a previous round of review and you feel that this manuscript is now acceptable for publication, you may indicate that here to bypass the “Comments to the Author” section, enter your conflict of interest statement in the “Confidential to Editor” section, and submit your "Accept" recommendation.

Reviewer #1: All comments have been addressed

2. Is the manuscript technically sound, and do the data support the conclusions?

Reviewer #1: Yes

3. Has the statistical analysis been performed appropriately and rigorously? 

Reviewer #1: Yes

4. Have the authors made all data underlying the findings in their manuscript fully available?

Reviewer #1: Yes

5. Is the manuscript presented in an intelligible fashion and written in standard English?

Reviewer #1: Yes

6. Review Comments to the Author

Reviewer #1: This revised version is suitable for publication. I think that this recent revised version will proper to academic societies.

7. PLOS authors have the option to publish the peer review history of their article (what does this mean?). If published, this will include your full peer review and any attached files.

Reviewer #1: No

---

## [Editor Report · Acceptance letter]

7 May 2024

PONE-D-24-02315R1 

PLOS ONE

Dear Dr. Tian, 

I'm pleased to inform you that your manuscript has been deemed suitable for publication in PLOS ONE. Congratulations! Your manuscript is now being handed over to our production team.

Kind regards, 

on behalf of

Dr. Kristofer Lasko 

Academic Editor

PLOS ONE